Maize-peanut rotational strip intercropping improves peanut growth and soil properties by optimizing microbial community diversity

Han Yi 1
Dong Qiqi 1
Zhang Kezhao 1
Sha Dejian 1
Jiang Chunji 1
Yang Xu 1
Liu Xibo 1
Zhang He 1
Wang Xiaoguang 1
Guo Feng 2
Zhang Zheng 2
Wan Shubo 2
Zhao Xinhua xinhua_zhao@syau.edu.cn 1
Yu Haiqiu yuhaiqiu@syau.edu.cn 1
1 College of Agronomy, Shenyang Agricultural University , Shenyang City , Liaoning Province , China
2 Shandong Academy of Agricultural Sciences , Jinan , Shandong , China
Khan Amanullah
Electronic publication date: 2022 Jul 28
Publication date: 2022
Volume: 10
Electronic Location ID: e13777
Received 2022 Apr 5; Accepted 2022 Jul 1
Copyright: ©2022 Han et al.
Copyright year: 2022
Copyright holder: Han et al.
License: This is an open access article distributed under the terms of the Creative Commons Attribution License, which permits unrestricted use, distribution, reproduction and adaptation in any medium and for any purpose provided that it is properly attributed. For attribution, the original author(s), title, publication source (PeerJ) and either DOI or URL of the article must be cited.
License URL: https://creativecommons.org/licenses/by/4.0/

Keywords: Rotational strip intercropping (RSI), Yield, Rhizosphere soil, Soil physical properties, Enzyme activities, 16S/ITS

Funding: The National Natural Science Foundation of China NO. U21A20217 This work was supported by Joint Funds of the National Natural Science Foundation of China (NO. U21A20217). The funders had no role in study design, data collection and analysis, decision to publish, or preparation of the manuscript.

==============================
Rotational strip intercropping (RSI) of cereals and legumes has been developed and widely carried out to alleviate continuous cropping obstacles, to control erosion and to improve field use efficiency. In this study, a four-year fixed-field experiment was carried out in northeast China with three treatments: continuous cropping of maize, continuous cropping of peanuts and rotational strip intercropping of maize and peanut. The results show that crop rotation improved the main-stem height, branch number, lateral branch length, and yield and quality of peanuts; the yield was the highest in 2018, when it was increased by 39.5%. RSI improved the contents of total N, available N, total P, available P, total K and available K; the content of available N was the highest in 2018, with an increase of 70%. Rhizosphere soil urease and catalase activities were significantly increased and were the highest in 2017, reaching 183.13% and 91.21%, respectively. According to a high-throughput sequencing analysis, the rhizosphere soil bacterial richness and specific OTUs decreased in peanut rhizosphere soil, while the fungal increased. There were differences in the bacterial and fungal community structures; specifically, the abundance of Acidobacteria and Planctomycetes increased among bacteria and the abundance of beneficial microorganisms such as Ascomycota increased among fungi. In conclusion, rotational strip intercropping of maize and peanut increased the yield and quality of peanuts and conducive to alleviating the obstacles facing the continuous cropping of peanuts. Among then, soil physicochemical properties, enzyme activity and microbial diversity were significantly affected the yield of peanut.

Introduction

In recent years, conflict, climate variability and extremes, economic slowdowns and downturns, and poverty and inequality have put the world off track from the goal of ending world hunger and malnutrition by 2030 (FAO, 2021a). Thus, to achieve the Sustainable Development Goals, the growing demand for crop production requires us to explore innovative ways to act in the context of the changing climate and loss of biodiversity (Morugán-Coronado et al., 2022). Diversified crop cultivation has shown benefits for soil macro- and microorganisms and improvement of crop yields; it is well known that intercropping and rotation are the essential operations of traditional agriculture and ever-present sustainable practices across different regional agroecosystems (Du et al., 2018; Li et al., 2020; Zheng et al., 2021). In China, rotational strip intercropping (RSI), a new cultivation pattern, has been widely applied to adapt to mechanized production, alleviate the degradation of field soil and ensure the safe production of grain and oil supplies (Zou et al., 2021). Compared with traditional intercropping or rotation, RSI is more conducive to mechanization, improving land-use efficiency, alleviating the conflict between cereal and oil crops for land and, more importantly, enriching the soil.

Beneficial interactions between cereal and legumes in strip intercropping have been reported, including high land-use efficiency, improvement of soil fertility, reduction in disease and pest incidence and ensuring of stable yield. An ecological feature of cereal–legume intercropping the improved utilization efficiency of natural resources, especially light. The photosynthetic active radiation (PAR) was significantly increased at the top of the soybean canopy and the photosynthetic rate (Pn) and radiation-use efficiency (RUE) of maize leaves close to the ear were significantly increased by optimizing maize row distance and gap width in maize–soybean strip intercropping (Liu et al., 2018; Liu et al., 2017). In addition, the interactions among roots in cereal and legume intercropping can promote nutritional efficiency, such as nitration (N) (Li et al., 2016) and use of phosphate (P) (Song et al., 2022), iron (Fe) (Dai et al., 2018) and zinc (Zn) (Xue et al., 2016), and reduce fertilizer application (Gong et al., 2021). Furthermore, pea–barley intercropping in rotation with durum wheat improved nitrogen-utilization efficiency and increased the overall sustainability of the rotation (Monti et al., 2019). Other key ecological functions of cereal–legume intercrops are reduced land-surface wind speed and soil erosion (Hombegowda et al., 2020). Maize (Zea mays L.) is one of the three most important cereal crops, while peanut (Arachis hypogaea L.) is the main oil crop and economic crop in the world. However, long-term continuous cropping has caused soil nutrient imbalance, serious soil erosion, bad seed quality and lower yield (FAO, 2021b; Mallano et al., 2021). Evidence indicates that intercropping with greater plant richness produces more roots, which contribute to the formation of soil aggregates; optimize the soil physical structure, which improves the soil chemistry parameters; and increase straw coverage, which decreases soil erosion (Gamboa et al., 2020; Zhang et al., 2020b). Remarkably, rotational strip intercropping of maize and peanut (RMP) increased soil carbon stocks, reduced soil erosion and carbon emission and improved soil characteristics (Zou et al., 2021). However, the underlying mechanisms of the effects of RSI on crop development and soil properties are still unclear.

Rhizosphere soil microorganisms play an important role in crop growth and can be used as effective biological indicators of soil conditions (Singh et al., 2009). It can clearly be seen that abnormal soil microflora is one of the main obstacles to continuous cropping (Mallano et al., 2021; Song et al., 2016; Yuan et al., 2021). Numerous previous studies have confirmed that crop intercropping and rotation are an effective way to break the continuous cropping barrier and an important part of conservation agriculture (Aschi et al., 2017; Hobbs, Sayre & Gupta, 2008). Among them, legume–grass rotation is an effective form of sustainable agriculture and an important measure to maintain soil biodiversity (Gwenambira-Mwika, Snapp & Chikowo, 2021; Zhao et al., 2021; Zou et al., 2021). Compared with continuous cropping of peanuts, the diversity and richness of bacteria were increased in peanut rhizosphere soil after maize–peanut rotation; the main bacterial phyla were Proteobacteria, Acidobacteria, Firmicutes, Bacteroidetes and Actinomycetes (Guo et al., 2021; Sudini et al., 2011). Other reports have indicated that continuous cropping significantly increased potential beneficial bacterial and fungal populations in soybean rhizosphere soil, including Pseudomonas, Nitrospira, Streptomyces and soybean cyst nematode (SCN) parasitic fungi, compared to short-term rotation (Song et al., 2016; Yuan et al., 2021). The order of crop planting and soil sampling time (Sudini et al., 2011), soil chemistry (Chodak et al., 2015) and rotation time (Liu et al., 2020) were the reasons for the differences in these studies. Researchers typically focus on specific fungi and bacteria when studying cropping obstacles, using whole-genome sequencing to uncover the molecular mechanisms of pathogenicity of plant pathogens at the gene level (Dou et al., 2008). Thus, understanding the roles and functions of bacteria and fungi in the rhizosphere of peanuts, as well as the integrated effects of rhizosphere microorganisms on crop performance, is fundamental and would contribute valuable information for agricultural production.

Under the premise of ensuring the maize planting area and increasing the legume planting area, the RSI pattern of maize and peanut has been developed and has become a very significant application prospect in northern China. However, previous studies have mainly focused on pattern selection, crop growth and development and variations in soil physical and chemical properties in maize/peanut intercropping, while limited knowledge exists of the microbial community changes in soil under RSI of maize and peanut (Gao et al., 2019; Li et al., 2019c; Wang et al., 2020; Zhang et al., 2020a). Therefore, we carried out RSI of maize and peanut, with 16 rows of maize and eight rows of peanut (16M/8P), to explore the variations of crop growth and microbial community in a 4-year fixed-point experiment. From the perspective of rhizosphere soil microorganisms, the objectives of this study were to (1) reveal the impact of RSI of maize and peanut on crop growth, soil properties and soil enzyme activities; (2) compare the differences in microbial functional diversity between continuous cropping and RSI of maize and peanut (3) explore the effect of bacterial and fungal correlations in rhizosphere soil on the yield and quality of the RSI system; and (4) determine whether the rhizosphere soil microbial community could be a key element in breaking the barriers that face continuous cropping.

Materials & Methods

Experiment site

The field experiment was conducted at Zhangwu Experimental Station (42°18′N 122°34′E) of Shenyang Agricultural University, northwest of Liaoning Province, China, from May 2015 to October 2018. The experimental site represents a typical semi-humid and semi-arid climate with an annual temperature of 7.2 °C and annual precipitation of 484.3 mm (Fig. S1). The annual effective accumulated temperature is 1644.8 °C, with a frost-free period of 156 d. The field soil is typical sandy loam, which was successively used for cropping of peanut for over 10 consecutive years. The nutritional characteristics of the topsoil (0–20 cm) in early spring of 2015 were as follows: 1.03 g kg−1 of organic matter, 65.2 mg kg−1 of available N, 12.2 mg kg−1 of available P, 76 mg kg−1 of available K and average pH 7.18.

Experimental design and crop management

A randomized complete block design with three replications was used in this experiment. Three planting patterns were adopted, including peanut continuous cropping (PC), maize continuous cropping (MC) and rotational strip intercropping of maize and peanut (RMP). The individual plot area was 600 m2 (12 m × 50 m) for both the continuous cropping and intercropping systems. Because the pre-season crop was peanut, both monoculture cropping and intercropping methods of peanut were continuous cropping. In the RMP system, 16 rows of maize intercropped with 8 rows of peanut (16M/8P) were established; maize and peanut were intercropped not only in wide strips but also every eight rows and were sequentially rotated inter- annually from 2015 to 2018. Each year, maize crop preceded peanut, following a strip rotation planting of maize under intercropping. Thus, the peanut strip returned to its original position every 4 years, achieving a cycle, whereas maize strips (eight rows) were planted with peanuts every 3 years (Fig. 1). The row spacing for all treatments was 50 cm in a south–north orientation, divided into 24 rows. The density of maize was 67,500 plants ha−1 with 29.6 cm of hill-spacing using single-ridge bare-field planting methods. However, peanut was sown at 300,000 plants ha−1 in single-seed staggered sowing with alternative sowing on small ridges with the narrow two-row method according to local habits, whereby the spacing between two lines was 10 cm and hill distance was 13.4 cm. The planting densities and patterns were the same for monoculture and intercropping.

Figure 1 Rotational strip intercropping of maize and peanut (RMP), peanut continuous cropping (PC) and maize continuous cropping (MC) planting pattern diagram.

Peanut (Nonghua13) and maize (Liangyu66) were selected in this study and the seeds were provided by Peanut Research Institute of Shenyang Agriculture University and Dandong Liangyu Seed Industry Co., Ltd, respectively. The two crops were sown on the same dates: specifically, on 18 May 2015, 15 May 2016, 20 May 2017, and 16 May 2018. The fertilizer for maize was applied at 368 kg N ha−1, 177 kg P2O5 ha−1 and 205 kg K2O ha−1, while the fertilizer for peanut was applied at 233 kg N ha−1, 267 kg P2O5 ha−1 and 250 kg K2O ha−1. Other management practices were conducted according to local agronomic practices.

Agronomic traits, yield, and quality of peanut

Three representative peanut samples were randomly selected from the center of each plot at the flowering and needle stages of peanuts in 2016, 2017 and 2018; agronomic traits such as plant height, branch number, lateral branch length and leaf area were measured, and the average values were calculated and then used for analysis. Yield samples were collected at the time of harvest each year. The length of the ridge was 1 m and three ridges were continuously used in each plot to measure peanut yield (Boote, 1982). The contents of oil and protein were determined with a near-infrared grain analyzer (INFRAJEC Nova; Flowserve Analytical Instruments, Denmark).

Nutrients and enzyme activities in peanut rhizosphere soil

Three representative peanut rhizosphere soil samples were collected in 2016–2018 during peanut flowering and needle stages, then mixed to form biological replicates, giving a total of three replicates. After removing the topsoil, we collected rhizosphere soil samples from 20 cm deep with a sterilized shovel, sieved them with a 20-mesh sieve to remove impurities such as plant roots and animal residues and distributed them into 15 mL sterile centrifuge tubes, with 10–15 g per tube (Guo et al., 2018).

Rhizosphere soil nutrients were collected as previously described in (Zhang et al., 2021). Total nitrogen (TN) was determined using the Kjeldahl method and the total phosphorus (TP) was determined using the Olsen method, while total potassium (TK) was determined using the flame photometry method. The soil available nitrogen (AN), available phosphorus (AP) and available potassium (AK) were determined using the alkaline hydrolysis diffusion method, molybdenum antimony colorimetry and flame photometry method, respectively.

Soil urease (UE) and catalase (CAT) activities were determined in accordance with the method described by Jin et al. (2009) (Jin et al., 2009).

DNA extraction and PCR amplification of 16S/ITS

In 2018, rhizosphere soil samples of continuous peanut (PC), rotation peanut (PR), continuous maize (CC) and rotational maize (CR) were stored at 280 °C for soil DNA extraction (Fig. 1); then, the Guangzhou Gene Denovo Biotechnology Co., Ltd. was commissioned to carry out microbial functional and structural diversity with BIOLOG ECO plates and high-throughput sequencing.

Data were collected as previously described in Yue et al. (2019) and Yang et al. (2019). Specifically, microbial DNA was extracted from soil samples using the E.Z.N.A. stool DNA Kit (Omega Biotek, Norcross, GA, US) according to the manufacturer’s protocols. The 16S rDNA V3–V4 region of the Eukaryotic ribosomal RNA gene was amplified by PCR (95 °C for 2 min, followed by 27 cycles at 98 °C for 10 s, 62 °C for 30 s and 68 °C for 30 s, with a final extension at 68 °C for 10 min) using primers 341F:CCTACGGGNGGCWGCAG and 806R:GGACTACHVGGGTATCTAAT, where the barcode was an eight-base sequence unique to each sample. The ITS region of the Eukaryotic ribosomal RNA gene was amplified by PCR (95 °C for 2 min, followed by 27 cycles at 98 °C for 10 s, 62 °C for 30 s and 68 °C for 30 s, with a final extension at 68 °C for 10 min) using primers ITS3_KYO2F 5′- GATGAAGAACGYAGYRAA -3′and ITS4R 5′-TCCTCCGCTTATTGATATGC -3′, where the barcode was an eight-base sequence unique to each sample .

The process of PCR reaction as previously described in Li et al. (2019b). PCR reactions were performed in triplicate in a 50 µL mixture containing 5 µL of 10 × KOD buffer, 5 µL of 2.5 mM dNTPs, 1.5 µL of each primer (5 µM), 1 µL of KOD polymerase and 100 ng of template DNA. Amplicons were extracted from 2% agarose gels and purified using the AxyPrep DNA Gel Extraction Kit (Axygen Biosciences, Union City, CA, US) according to the manufacturer’s instructions, and quantified using Quanti Fluor -ST (Promega, Madison, WI, USA). Purified amplicons were pooled in equimolar ratios and paired-end sequenced (2 × 250) on an Illumina platform according to the standard protocols. The raw reads were deposited into the NCBI Sequence Read Archive (SRA) database: accession number PRJNA810273 for bacteria and PRJNA810272 for fungi.

Statistical and bioinformatic analysis

Peanut agronomic traits, yield and quality, rhizosphere soil nutrients and soil enzyme activities were tested for differences among cropping methods with the Duncan’s test one-way analysis of variance (ANOVA) using SPSS 23.0 for Windows (IBM SPSS Inc., USA). Different years and planting pattern and year x planting pattern significant differences were defined at *, p < 0.05 and **, p < 0.01. Using Origin Pro version 9.0 (Origin Lab Corporation, Northampton, MA, United States) and R software (version 4.0.3) for drawing.

Bioinformatic analysis of bacteria and fungi consistent with previously described in Wang et al. (2022). Specifically, Chao1 and Shannon were calculated in QIIME. The statistics of group Alpha index comparison were calculated using a Welch’s t-test and a Wilcoxon rank test in R.

The effective tags were clustered into operational taxonomic units (OTUs) of ≥97% similarity using the UPARSE (Edgar, 2013) pipeline. The tag sequence with the highest abundance was selected as the representative sequence within each cluster. An Upset analysis between groups was performed in R to identify specific and common OTUs.

The representative sequences were classified into organisms by a naive Bayesian model using RDP classifier (Version 2.2) (Wang et al., 2007) based on the SILVA (Pruesse et al., 2007) Database (https://www.arb-silva.de/) and UNITE (Koljalg et al., 2005) Database (https://unite.ut.ee/). The abundance statistics of each taxonomy and phylogenetic tree were constructed in a Perl script and visualized using SVG.

Network analysis of bacterial and fungal community correlations were carried out by using OmicShare Tools platform (http://www.omicshare.com/tools).

The functional capacity of the microbial community and functional categorization were based on the Kyoto Encyclopedia of Genes and Genomes (KEGG) pathways (Kanehisa, 2019; Kanehisa et al., 2021; Kanehisa & Goto, 2000). The functional group (guild) of the OTUs was inferred using Tax4 Fun (v1.0) (Asshauer et al., 2015) and FUN Guild (v1.0) (Nguyen et al., 2016).

Results

Effects of rotational strip intercropping in different years on peanut agronomic characteristics

Year and planting pattern had (p < 0.01) significant effects on peanut agronomic traits and year x planting pattern showed no significance. The results of the study (Table 1) show that peanut main-stem height, branch number and side branch length with PR were significantly higher than with PC, account for 14.89%, 10.64% and 13.05%, respectively. With the increase in crop rotation years, the main-stem height and lateral branch length gradually decreased, becoming significant in 2018. The number of branches and leaf area increased first and then decreased and the leaf area changed significantly between years. So that, reasonable rotational strip intercropping significantly changed the agronomic characteristics of peanuts and promoted the growth of peanuts.

Table 1 Effects of rotational strip intercropping in different years on agronomic traits of peanut.

Year	Planting pattern	Main stem height (cm)	The number of branches	Lateral branch length (cm)	Leaf area (cm2)	
2016	PR	34.90 ± 1.60a	5.60 ± 0.10a	36.07 ± 0.90a	2051.20 ± 114.48a	
	PC	30.43 ± 0.93b	5.07 ± 0.06b	31.70 ± 1.25b	1908.03 ± 18.42a	
2017	PR	32.53 ± 0.80a	5.93 ± 0.25a	32.60 ± 6.15a	2285.17 ± 33.97a	
	PC	29.30 ± 1.37b	5.17 ± 0.15b	26.63 ± 0.42a	2101.57 ± 23.72b	
2018	PR	32.63 ± 1.42a	5.37 ± 0.15a	29.47 ± 0.81a	1795.63 ± 38.34a	
	PC	27.43 ± 1.07b	5.03 ± 0.12b	26.43 ± 1.11a	1654.87 ± 10.87b	
Year	*	*	**	**	
Planting pattern	**	**	**	**	
Year × Planting pattern	NS	NS	NS	NS	
Notes.

Different lowercase letters indicate significant differences in peanut agronomic traits between different planting methods in the same year (p < 0.05); * and ** indicate significant (p < 0.05) and extremely significant (p < 0.01) differences; NS means not significant. PR, rotation peanut; PC, continuous peanut.

Effects of rotational strip intercropping in different years on peanut yield and quality

Year, planting pattern and year x planting pattern had significant effects on peanut yield and oil content. The results of the study (Table 2) show that peanut yield with PR was significantly (p < 0.05) higher than that with PC. Multiyear continuous cropping significantly reduced the yield of peanut. Compared with PC, the yield of peanut increased the most in 2018, accounting for 39.5%. The contents of oil and protein with PR were significantly (p < 0.05) higher than with PC. With the increase in the rotation years, the content of oil significantly decreased, and the content of protein increased and reached a peak in 2018. So that, reasonable rotational strip intercropping increased the peanut yield and content of protein.

Table 2 Effects of rotational strip intercropping in different years on yield and quality of peanut.

Year	Planting pattern	Yield (kg hm−2)	○	Content of oil (%)	Protein (%)	
2016	PR	6548.00 ± 283.50a	18.17%	47.24 ± 0.09a	73.23 ± 0.13a	
	PC	5551.00 ± 209.21b		46.01 ± 0.09b	72.29 ± 0.06b	
2017	PR	6058.00 ± 107.40a	28.80%	46.90 ± 0.17a	73.49 ± 0.04a	
	PC	4706.50 ± 177.10b		45.54 ± 0.10b	72.78 ± 0.07b	
2018	PR	6263.00 ± 101.13a	39.50%	45.24 ± 0.08a	76.65 ± 0.09a	
	PC	4492.50 ± 181.31b		45.83 ± 0.08b	75.56 ± 0.06b	
Year	**		**	**	
Planting pattern		**		**	**	
Year  ×  Planting pattern		*		**	NS	
Notes.

Different lowercase letters indicate significant differences in peanut yield and quality between different planting methods in the same year (p < 0.05); * and ** indicate significant (p < 0.05) and extremely significant (p < 0.01) differences; NS means not significant. PR, rotation peanut; PC, continuous peanut; ○, Yield (RP-CP)/CP*100%.

Effects of rotational strip intercropping in different years on nutrients and enzyme activities of peanut rhizosphere soil

Year, planting pattern and year x planting pattern had significant effects on rhizosphere soil available N (AN), total P (TP) and available P (AP) (Table 3). The soil nutrients of PR significantly increased over the three years (2016–2018). After rotational strip intercropping, the soil total P (TP) and available P (AP) reached their highest levels in 2017. The soil total N (TN), total P (TP) and available P (AP) contents significantly decreased with the increase in the rotation years. The contents of soil available N (AN) first decreased and then increased, while the contents of soil total K (TK) and available K (AK) first increased and then decreased.

Table 3 Effects of rotational strip intercropping in different years on soil nutrient content in peanut.

Year	Planting pattern	TN(g kg−1)	AN(mg kg−1)	TP(g kg−1)	AP(mg kg−1)	TK(g kg−1)	AK(mg kg−1)	
2016	PR	0.58 ± 0.02a	37.41 ± 0.28a	0.04 ± 0.002a	15.65 ± 0.41a	5.72 ± 0.43a	4.99 ± 0.12a	
	PC	0.47 ± 0.01b	34.42 ± 0.25b	0.03 ± 0.001b	11.86 ± 0.26b	5.31 ± 0.23a	4.78 ± 0.14a	
2017	PR	0.57 ± 0.02a	36.37 ± 0.27a	0.04 ± 0.001a	13.66 ± 0.30a	5.79 ± 0.14a	5.54 ± 0.16a	
	PC	0.43 ± 0.01b	28.21 ± 0.31b	0.02 ± 0.001b	8.18 ± 0.49b	4.66 ± 0.24b	5.19 ± 0.10b	
2018	PR	0.50 ± 0.02a	38.85 ± 0.09a	0.03 ± 0.001a	13.55 ± 0.52a	5.73 ± 0.23a	5.45 ± 0.32a	
	PC	0.41 ± 0.01b	22.79 ± 0.43b	0.02 ± 0.002b	7.27 ± 0.32b	4.59 ± 0.25b	5.14 ± 0.22a	
Year	*	*	*	*	NS	*	
Planting pattern	*	*	*	*	*	NS	
Year  ×  Planting pattern	NS	*	*	*	NS	NS	
Notes.

Different lowercase letters indicate significant differences in peanut soil nutrient content between different planting methods in the same year (p < 0.05); * and ** indicate significant (p < 0.05) and extremely significant (p < 0.01) differences; NS means not significant. PR, rotation peanut; PC, continuous peanut; TN, total N; AN, available N; TP, total P; AP, available P; TK, total K; AK, available K.

The planting pattern had a significant effect on soil urease and catalase activities (Fig. 2). The soil urease and catalase activities significantly increased after crop rotation. Compared with PC, soil activity increased by 86.1%, 203.48% and 183.13% in 2016, 2017 and 2018, respectively. Soil catalase activity increased by 75.92%, 91.21% and 78.62% in 2016, 2017 and 2018, respectively. Soil urease and catalase activities reached their highest values in 2017.

Figure 2 Effects of rotational strip intercropping in different years on soil enzyme activity of peanut.

(A) Soil urease activity; (B) soil catalase activity. Different lowercase letters indicate significant differences in peanut rhizosphere soil between different planting methods in the same year (p < 0.05); asterisks (* and **) indicate significant (p < 0.05) and extremely significant (p < 0.01) differences; NS means not significant. PR, rotation peanut; PC, continuous peanut.

Effects of rotational strip intercropping in different years on microbial diversity in peanut rhizosphere soil

The rarefaction curve and rank abundance curve showed that the sequencing depth covered all the taxa in the sample, the distribution was uniform, and the sequencing results were stable and reliable (Fig. S2). In the bacterial community, compared with PC, the chao1 and Shannon indexes were decreased in PR (Figs. 3A, 3B). Compared with CC, the chao1 index was increased in CR and the Shannon index was decreased in CR (Figs. 3A, 3B). These results showed that crop rotation reduced the richness and diversity of bacterial community in peanut rhizosphere soil. In the fungal community, compared with PC, the chao1 and Shannon indexes were increased in PR (Figs. 3C, 3D). Compared with CC, the chao1 and Shannon indexes were decreased in CR (Figs. 3C, 3D). These results showed that crop rotation increased the richness and diversity of fungal community in peanut rhizosphere soil.

Figure 3 Effects of rotational strip intercropping in different years on rhizosphere soil microbial diversity indexes.

(A) Chao1 index of bacteria; (B) Shannon index of fungi; (C) Chao1 index of fungi; (D) Shannon index of fungi. PR, rotation peanut; PC, continuous peanut; CR, rotation maize; CC, continuous maize.

Bacterial and fungal diversity were negatively correlated (r =−0.3555; p = 0.2567) (Fig. 4A), indicating that the diversity of bacterial and fungal communities is different under different planting patterns. At the OTU level, bacterial community structural changes were significantly correlated with those of the fungal community (p = 0.001), suggesting that bacterial and fungal structural changes have different response-driven mechanisms (r = 0.607) (Fig. 4B).

Figure 4 Correlation scatter plot of 16S and ITS alpha diversity with Shannon index (A) and beta diversity correlations in OTU levels (B).

Effects of rotational strip intercropping in different years on OTUs in rhizosphere soil

An OTU (operational taxonomic unit) is an artificial taxon which contributes to the classification of soil microorganisms based on sequencing. In the bacterial community, the shared OTUs were 400 in PR and PC. The specific OTUs were 616 in PR and 1519 in PC (Fig. 5A). It was shown that rotation with maize reduced the specific bacterial OTU number in the peanut rhizosphere. The sum of total observed OTUs in bacteria was 1,452, the shared OTUs were 164 in CR and CC and the specific OTUs were 331 in the CR and 957 in CC. In the fungal community, the shared OTUs were 32 in PR and PC. The specific OTUs were 136 in the PR and 111 in PC (Fig. 5B). In contrast to bacteria, rotation with maize increased the specific fungal OTU number in the peanut rhizosphere. The sum of total observed OTUs in fungi was 382, and the shared OTUs were 46 in CR and CC. The specific OTUs were 144 in the CR and 192 in CC.

Figure 5 Upset diagrams of bacteria (A) and fungi (B) community.

PR, rotation peanut; PC, continuous peanut; CR, rotation maize; CC, continuous maize.

Effects of rotational strip intercropping in different years on the microbial community in rhizosphere soil

At the phylum level, the top 10 phyla in the bacterial community were found through sequence classification. Actinobacteria, Proteobacteria, Planctomycetes and Acidobacteria were the dominant phyla and accounted for more than 10% of the total biodiversity in each sample (Fig. 6A; Table S1). Compared with PC, the relative abundances of Planctomycetes and Acidobacteria in PR were increased by 24.32% and 4.72%, respectively. Compared with PC, the relative abundances of Actinobacteria and Proteobacteria were decreased by 4.75% and 2.73%, respectively. In the fungal community, Ascomycota were the dominant fungi and accounted for more than 60% of the total biodiversity in each sample (Fig. 6B; Table S1). Compared with PC, the relative abundances of Ascomycota and Mortierellomycota in PR were increased by 22.55% and 38.10%, respectively. The relative abundances of Basidiomycota, Glomeromycota and Chytridiomycota in PR were decreased by 23.42%, 31.31% and 67.68%, respectively. Through KEGG functional prediction, it was found that the fructose and mannose metabolism and methane metabolism pathways involved in the bacterial community in the rhizosphere soil of PR were higher than in PC (Fig. S3A). The fungus functional annotation discovery by FUNGuild showed that saprophytic fungi had a higher relative abundance in PR (Fig. S3B).

Figure 6 Effects of rotational strip intercropping in different years on rhizosphere soil microbial community.

(A) Bacterial community; (B) fungal community. PR, rotation peanut; PC, continuous peanut; CR, rotation maize; CC, continuous maize.

Actinobacteria were extremely significantly positively correlated with Glomeromycota (r = 0.99; p = 0.005). Proteobacteria were significantly positively correlated with Glomeromycota (r = 0.98; p = 0.011). Acidobacteria (r = −0.96; p = 0.034) and Verrucomicrobia (r = −0.98; p = 0.015) were significantly negatively correlated with Glomeromycota. Ascomycota were negatively correlated with Gemmatimonadetes (r = −0.82; p = 0.178) and Firmicutes (r = −0.81; p = 0.188) (Fig. 7).

Figure 7 Network analysis of 16S and ITS.

Discussion

The study shows that rotational strip intercropping of maize and peanut increased the main-stem height, branch number, side branch length, yield, and quality of peanut (Tables 1, 2). This is consistent with the results reported by Zou, who concluded that the yield of peanut in maize–peanut rotation was 30%–35% higher than that of continuous cropping of peanuts in 2017 and 2018 (Zou et al., 2021). In addition, in the study of peanut with wheat and cotton rotation, it was also found that the yield of peanut was increased after the rotation (Chi et al., 2019; Liu et al., 2019). Based on the results of previous studies, this study also confirmed that the rotational strip intercropping system has a positive impact on continuous cropping obstacles such as improved agronomics traits of peanut, and effectively increased the yield and quality.

The results of this study show that the nutrient content of peanut rhizosphere soil was significantly higher than that after continuous cropping of peanut in a 4-year field experiment of rotational strip intercropping of maize and peanut. Moreover, the content of available N in peanut rhizosphere soil reached the highest level in 2018 (Table 3). The rotation of leguminous and gramineous crops could effectively improve the nutrient supply and soil fertility of continuous cropping soil, especially the supply of soil nitrogen. A study showed that legume crops through biological nitrogen fixation and rhizosphere nitrogen deposition provided nitrogen sources for non-legume crops in crop rotation, which is crucial for the sustainable development of legume-based crop rotation systems (Zou et al., 2021). In addition, during the long-term complex crop rotation, the activity of soil hydrolase and the content of mineral-associated organic matter nitrogen were increased, suggesting that the complexity of long-term crop rotation affects microbial nitrogen cycling and availability through feedbacks on plant physiology (Bowles et al., 2022). Soil enzyme activity is an important indicator of soil fertility, soil quality and soil health. It can represent the ability of material metabolism in soil and it can also reflect the absorption and utilization of nutrients by crops and growth conditions (Adetunji et al., 2017). Crop diversity in a rotation system affects the quality and quantity of residues, which also provide different polyphenolic compounds that affect the release of available nitrogen and modulate soil biology (Wang et al., 2011; Zibilske & Bradford, 2007). In other research studies with crop rotation of Gramineae and legumes, it was found that crop rotation significantly increased the activities of C, N and P cycle-related enzymes in soil, such as β-glucosidase, urease, catalase, phosphatase etc., whose activities were increased (Alhameid et al., 2019; Aschi et al., 2017; Malobane et al., 2020). Consistent with previous studies, the present study found that the activities of urease and catalase in the peanut rhizosphere soil were significantly increased after maize and peanut rotation (Fig. 2). The reason for this phenomenon is that crop diversity had a greater positive effect on soil physicochemical properties and biological properties (Gunasekaran, Kaliappan & P, 2021). It was further confirmed that, compared with continuous cropping, rotational strip intercropping of maize and peanut promoted plant uptake of soil nutrients.

The root absorbs nutrients and water, so any changes in the rhizosphere environment has an impact on plant growth. Plants actively shape their rhizosphere community, which, in turn, profoundly alters plant growth (Philippot et al., 2013). The peanut flowering and needle stage is the most critical period for accumulated yield (Sudini et al., 2015). The crop rotational system is considered a promising management practice to improve plant nutrition by affecting biological and chemical processes in the rhizosphere to promote nutrient uptake (especially Zn, P and K) (Inal et al., 2007). Crop rotation affects soil microbial communities and soil quality. The present study shows that the bacterial richness and specific OTUs decreased and fungal richness and specific OTUs increased in peanut rhizosphere soil, compared with what was obtained with the continuous cropping of peanut (Figs. 3, 5). Previous studies have shown that soil microbial biomass was larger after crop rotation and presented more beneficial microorganisms than continuous cropping, such as Proteobacteria, Bacteroidetes, Firmicutes, Acidobacteria, Actinomycetes, etc. (Lupwayi et al., 2021; Nayyar et al., 2009; Tang et al., 2020; Wang et al., 2015; Yuan et al., 2021). This is consistent with the results of this study, whereby the relative abundance of Actinobacteria and Ascomycota was the highest in the rhizosphere soil of rotational strip intercropping of maize and peanut (Fig. 6). Therefore, our results indicate that the changes in the relative abundance of bacterial and fungal between the rotational strip intercropping of maize and peanut and continuous cropping treatments further verified the differences in the bacterial and fungal community structures (Fig. 4B). The reasons for the differences may be related to changes in the soil physicochemical properties caused by root exudates, such as organic acids, phenolics and flavonoids. In the future, the effects of the selection and adaptation mechanism of the composition and quantity of root exudates on rhizosphere soil microorganisms should be further explored (Sudini et al., 2015).

It has been reported that Actinobacteria are important to C cycling in soil because they contain enzymes capable of degrading cellulose (Lupwayi et al., 2021). Planctomycetes were among the widespread and abundant bacterial communities in soil which are involved in the degradation of organic matter. The related bacterial genera could decompose chitin and play a role in carbohydrate metabolism (Buckley et al., 2006; Ivanova et al., 2018). Soil fungi are responsible for a range of important ecological functions, such as affecting carbon sequestration through plant life and nutrient mineralization (Landinez-Torres et al., 2020). Ascomycota are the most typical fungi in PR, playing an important role in organic matter decomposition (Yelle et al., 2008). Through the 16S+ITS association analysis, it was found that there were complex correlations between bacterial and fungal communities (Fig. 7), consistently with previous studies (Jiang et al., 2020; Li et al., 2019a). Glomeromycota were the core fungi and there was a significant correlation with bacteria, which also indicated that Glomeromycota significantly promoted the increase in the dominant bacteria Acidobacteria in the rhizosphere soil of rotation-cropping peanut. Therefore, the bacterial and fungal communities were closely related and Glomeromycota led to an increase in the relative abundance of Acidobacteria, which led to functional changes in the bacterial and fungal communities, suggesting different response-driving mechanisms for bacteria and fungi.

Conclusions

Four years of field experiments showed that rotational strip intercropping of maize and peanut could improve yield and quality of peanuts and could alleviate the obstacles to continuous cropping of peanuts. The main-stem height, branch number and side branch length of peanut were increased in rotational strip intercropping. The content of available N was the highest, followed by significantly increased activities of urease and catalase in peanut rhizosphere soil, which catalyzed soil organic matter decomposition and nutrient cycling. Rotational strip intercropping changed the structure of the soil microbial community and the bacterial and fungal communities promoted each other, increasing the abundance of beneficial microorganisms, such as Acidobacteria, Planctomycetes, Ascomycota, etc., in peanut rhizosphere soil. The interaction of maize and peanut optimized the structure of the peanut rhizosphere bacterial and fungal community in the rotational strip intercropping, improved soil nutrient distribution, and promoted peanut growth and yield.

Supplemental Information

Supplemental Information 1 Raw data for soil urease & catalase activity, agronomic traits, yield and quality of peanuts, and soil nutrient content

Three representative peanut samples were randomly selected from the center of each plot at the flowering and needle stages of peanuts in 2016, 2017 and 2018. PR, rotation peanut; PC, continuous peanut.

Click here for additional data file.

Supplemental Information 2 Maize-peanut rotational strip intercropping improves peanut growth and soil properties by optimizing microbial community diversity

This dataset contains additional material for the article’s average temperature, rainfall, and rhizosphere soil microbe-related indicators.

Click here for additional data file.

We thanks to Associate Professor Xining Gao for providing us with the meteorological data from 2015 to 2018. We would also like to Pei Guo for drawing the planting pattern diagram.

Additional Information and Declarations

Competing Interests

Author Contributions

Data Availability

The authors declare there are no competing interests.

Yi Han conceived and designed the experiments, analyzed the data, prepared figures and/or tables, and approved the final draft.

Qiqi Dong conceived and designed the experiments, prepared figures and/or tables, and approved the final draft.

Kezhao Zhang conceived and designed the experiments, prepared figures and/or tables, and approved the final draft.

Dejian Sha conceived and designed the experiments, prepared figures and/or tables, and approved the final draft.

Chunji Jiang conceived and designed the experiments, authored or reviewed drafts of the article, and approved the final draft.

Xu Yang performed the experiments, analyzed the data, authored or reviewed drafts of the article, and approved the final draft.

Xibo Liu performed the experiments, analyzed the data, authored or reviewed drafts of the article, and approved the final draft.

He Zhang performed the experiments, analyzed the data, authored or reviewed drafts of the article, and approved the final draft.

Xiaoguang Wang performed the experiments, analyzed the data, authored or reviewed drafts of the article, and approved the final draft.

Feng Guo performed the experiments, authored or reviewed drafts of the article, and approved the final draft.

Zheng Zhang performed the experiments, authored or reviewed drafts of the article, and approved the final draft.

Shubo Wan performed the experiments, authored or reviewed drafts of the article, and approved the final draft.

Xinhua Zhao conceived and designed the experiments, prepared figures and/or tables, and approved the final draft.

Haiqiu Yu conceived and designed the experiments, prepared figures and/or tables, and approved the final draft.

The following information was supplied regarding data availability:

The raw reads are available at the NCBI Sequence Read Archive (SRA) database: PRJNA810273 (bacteria) and PRJNA810272 (fungi).

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
