# Peer review of "Maize-peanut rotational strip intercropping improves peanut growth and soil properties by optimizing microbial community diversity"

_PeerJ, doi:10.7717/peerj.13777_

## Round 0.1 · original submission · Major Revisions

The manuscript needs major revision.

Reviewer 1 ·

Basic reporting

no comment

Experimental design

no comment

Validity of the findings

no comment

Additional comments

The authors collected extensive data through years of field trial investigations of strip rotational intercropping of maize and peanut. And the logic of the manuscript is clear, especially the correlation analysis of bacterial and fungal communities in rhizosphere soil is necessary. However, the section on the results needs to be supplemented with specific values. Also, authors need to check the text format of the manuscript and correct it before accepting it.
Line220: What is the significance P value please? Below is the same.
Line 221-222, Line 230-231, Line232-233: Please specify the increased or decreased value.
Line260, Line263:chang the P to p
Line295-295: The analysis of bacterial and fungal correlations is novel and necessary, and we suggest that the analysis in this part should be more specific, using numerical values to illustrate.
Line300-308: We recommend that the reasons for analyzing yield increases should be discussed in detail, eg, agronomic traits can also have an impact on yield.
Line337: flowering and needle stages
Line375-383: Conclusion Part of the logical relationship is scattered, we suggest rearranging, such as rotational strip intercropping of maize and peanut was increased yield, and then separately explain the reasons for the increase in yield.

Reviewer 2 ·

Basic reporting

The submitted manuscript aims to understand the effect of rotational strip intercropping of maize and peanut on soil microbial community and crop growth. The manuscript is clearly written, well structured and presents interesting results. I have minor comments below:

- Overall throughout, authors have often interpreted correlation as causation. Please check these statements in the manuscript (for example Line 28 - 30 in the abstract) and rewrite.

- Could the authors please include full forms at the beginning of each section when they first mention a terminology such as PR or PC.

- Line 229 & 239: if there is a significant interaction, the authors must first state what this overall means for that particular result.

- ‘Reasonably’ used in lines 235 and 226 could be replaced with another more relevant word/words.

- Line 256-259: In addition to mentioning that Shannon index increased, authors should also mention what this means for a specific variable to improve the readibility of the results.

- Line 260 – 262: Negative correlation does not indicate the opposite effects of different treatments. This should be rephrased.

Experimental design

- Experimental design and statistical method is unclear. Authors should provide clearer description of variables included in the statistics? Was cropping method the only variable. Did they include year or cropping cycle or Block in the statistical analysis? Were there any random effects?

- Figure legend description does not match the figure in Figure 1

Validity of the findings

The conclusion and end of discussion section are more like a repetition of results. The authors could also mention what is the relevance of their findings and potentially discuss how the results relate back to the origina research questions. The authors could also discuss the limitations of their work.

---

## Round 0.2 · accepted · Accept

The manuscript is improved and accepted for publication, thanks